# Numerical Prediction of the Fatigue Crack Growth Rate in SLM Ti-6Al-4V Based on Crack Tip Plastic Strain

**Fábio F. Ferreira, Diogo M. Neto, Joel S. Jesus, Pedro A. Prates and Fernando V. Antunes ***

Univ Coimbra, CEMMPRE, Department of Mechanical Engineering, Rua Luís Reis Santos, Pinhal de Marrocos, 3030-788 Coimbra, Portugal; uc2015260022@student.uc.pt (F.F.F.); diogo.neto@dem.uc.pt (D.M.N.); joel.jesus@uc.pt (J.S.J.); pedro.prates@dem.uc.pt (P.A.P.)

* Correspondence: fernando.ventura@dem.uc.pt; Tel.: +351-239790700

**Abstract:** This study presents a numerical model to predict the fatigue crack growth (FCG) rate in compact tension specimens under constant amplitude cyclic loadings. The material studied is the Ti-6Al-4V titanium alloy produced by selective laser melting, which was submitted to two different post-treatments: (i) hot isostatic pressing, and (ii) heat treatment. The developed finite element model uses the cumulative plastic strain at the crack tip to define the nodal release. Two different FCG criteria are presented, namely the incremental plastic strain (IPS) criterion and the total plastic strain (TPS) criterion. The calibration of the elasto-plastic constitutive model was carried out using experimental data from low cycle fatigue tests of smooth specimens. For both proposed crack growth criteria, the predicted *da/dN-ΔK* curve is approximately linear in log-log scale. However, the slope of the curve is higher using the TPS criterion. The numerical predictions of the crack growth rate are in good agreement with the experimental results, which indicates that cyclic plastic deformation is the main damage mechanism. The numerical results showed that increasing the stress ratio leads to a shift up of the *da/dN-ΔK* curve. The effect of stress ratio was dissociated from variations of cyclic plastic deformation, and an extrinsic mechanism, i.e., crack closure phenomenon, was found to be the cause.

**Keywords:** fatigue crack growth; finite element analysis; plastic strain; Ti-6Al-4V; titanium alloy; additive manufacturing

---

## 1. Introduction

Additive manufacturing (AM) technology enables the production of complex lightweight structures and tailor-made products with minimum tooling and low material waste. Therefore, these technologies have an increasing importance in different fields, namely in aircraft and medical industries. One of the most widespread AM processes is selective laser melting (SLM), where a laser source is used to fuse a metallic powder. It is an automated technique for fabricating complex metal parts directly from computer aided design (CAD) data by fusing metal powders. It does not need to remove any binder and, theoretically, any castable materials can be used in the SLM process, although careful selection of process conditions is essential for successful operation. However, the high thermal gradient induced by rapid solidification after laser melting may cause residual stresses and cracks. Typical defects are pores produced by initial powder contamination, evaporation or local voids after powder-layer deposition and lack of fusion defects. Defectiveness is a major reason of concern for the fatigue performance of SLM parts. For this purpose, efforts are made both from the technological standpoint, mainly aimed at optimizing the SLM process parameters and defining suitable post-sintering thermo-mechanical

treatments. Heat treatment, such as annealing and hot isostatic pressing, can improve the final properties of the parts by stress relieving and full densification, respectively [1].

Titanium alloys, and Ti-6Al-4V in particular, are widely used in aerospace and medical industries. In fact, these materials present high strength-to-weight ratio, good corrosion resistance, fatigue resistance up to 300 °C, excellent thermal stability, and biocompatibility [2–4]. The Ti-6Al-4V alloy is a two-phase titanium alloy consisting of a hexagonal close-packed $\alpha$-phase and a body-centered cubic $\beta$-phase [5]. Manufacturing of titanium components by conventional machining processes is difficult and costly due to several inherent properties, such as the low thermal conductivity and high chemical reactivity with many cutting tool materials, which can lead to premature failure of the machining tools and reduced productivity [6,7]. The widespread use of Ti alloys is also limited because of difficulties in cost, and low recyclability [8]. The high reactivity of titanium to interstitial elements, such as oxygen, carbon, nitrogen, and hydrogen, in the molten state is the main obstacle to laser processing of titanium. Considerable efforts have also been made to devise suitable post-sintering thermo-mechanical treatments able to bring the fatigue properties of Ti-alloys to a level comparable with that of wrought material conditions. The most successful of them consists in combining hot isostatic pressing (HIP) and mechanical machining to confer to the part its final shape. In this way, it is possible to remove, or at least diminish, internal defectiveness and decrease the surface roughness. The HIP treatment involves simultaneous application of high pressure to a Ti-6Al-4V component under high temperature conditions in an inert gas atmosphere. The high pressure applied causes closure of pores by low-scale plastic flow and material transfer, which, under optimal conditions, may also bond the pore interfaces [9]. HIP also produces significant microstructural coarsening, resulting from self-diffusion, with grain growth by almost one order of magnitude [10].

In components submitted to cyclic loading, design against fatigue is fundamental. The presence of defects reduces the initiation life, and increase the importance of propagation life. Therefore, the damage tolerance approach, which assumes the presence of defects in components, is particularly adequate to study fatigue life in components produced by additive manufacturing. Sedmak et al. [4] predicted the fatigue life of a hip joint prosthesis using XFEM and assuming $\Delta K$ as the crack driving force. Paris' law was obtained experimentally using three-point bending specimens. Oguma and Nakamura [11] and Yoshinaka et al. [12] compared fatigue crack growth (FCG) results in air and vacuum. A great influence of environment was found, particularly at low values of $\Delta K$. In fact, $\Delta K$ has always been assumed to be the crack driving force in the study of Ti-6Al-4V alloy [11–13]. This seems logical since Ti-6Al-4V is a high strength material, however, as will be seen later in this paper, the validity of the small-scale yielding assumption is not straightforward. The use of non-linear parameters as the crack driving force is an interesting alternative because the emphasis is placed on the irreversible and non-linear phenomena responsible for FCG. In fact, these parameters provide a better understanding of crack tip phenomena and include crack closure in a natural way. A linear correlation was found between the plastic crack tip opening displacement (CTOD), determined experimentally [14] or numerically [15], and *da/dN*. This approach was used to predict the effect of material properties [16] and notch geometry [17], among other aspects. An alternative approach, based on cumulative plastic strain measured immediately of crack tip, was applied successfully to predict the FCG rate in the 2024-T351 aluminium alloy [18].

The main objective of this research is to study FCG in the Ti-6Al-4V alloy. Compact tension (CT) and cylindrical specimens were produced by SLM and posteriorly submitted to post-treatments of HIP or heat treatment. The CT specimens were tested in order to obtain classical da/dN-ΔK curves. A parallel numerical analysis, replicating the experimental FCG analysis, was developed assuming that crack tip plastic deformation is the driving force, in order to have a better understanding of crack tip phenomena. The elasto-plastic behavior was modeled using stress-strain loops obtained experimentally in the uncracked cylindrical specimens. FCG rates were predicted assuming that node release occurs when a critical value of accumulated plastic strain is reached. This critical value was calibrated using one experimental value of *da/dN*-ΔK curve. The comparison of the numerical FCG

rates with the experimental ones allows to validate the proposed FCG criterion, as well as highlight the importance of the boundary conditions adopted in the numerical model.

## 2. Experimental Work

### 2.1. Material and Specimens

Figure 1 shows the two specimen geometries used in the experimental tests. Figure 1a presents the geometry used in the low cycle fatigue tests and tensile tests, defined according to ASTM E606 [19], while Figure 1b shows the specimens used in FCG tests, defined according to ASTM E647 [20]. All specimens were produced by SLM process, layer by layer in a reverse mesh at 45°, using a 3D Systems model ProX DMP 320 (3D Systems, Rock Hill, SC, USA). Metal powder of Ti-6Al-4V (grade 23) titanium alloy, with the chemical composition indicated in Table 1 (given by the manufacturer), was used. The energy density used to fabricate the specimens was 57 J/mm$^3$ and the thickness of each layer was 30 μm. The deposition direction of layers is indicated in Figure 1.

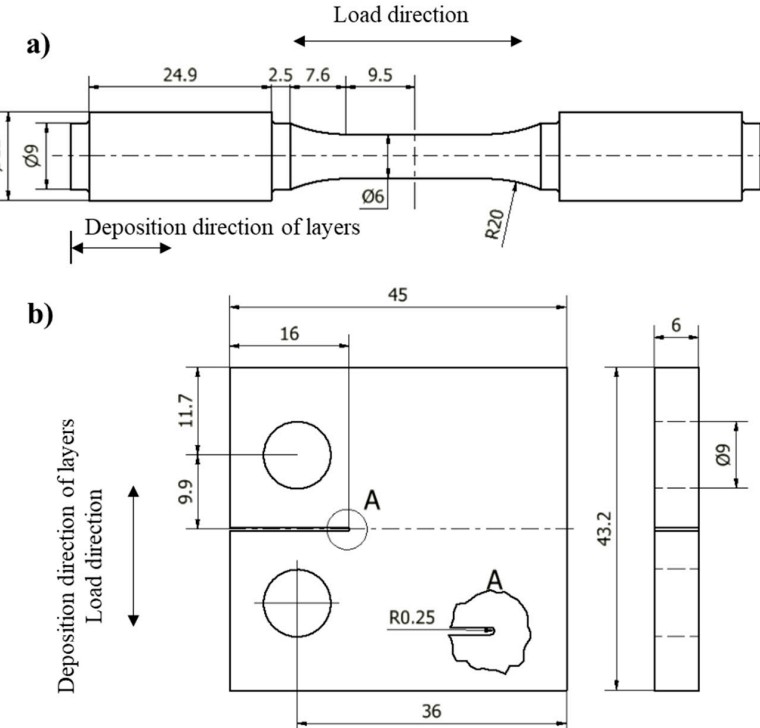

**Figure 1.** Specimens' geometries (mm): (**a**) Low cycle fatigue tests; (**b**) fatigue crack growth tests.

**Table 1.** Chemical composition of the Ti-6Al-4V titanium alloy powder (wt%).

| Al | He | Fe | Y | C | V | O | N | Ti |
|---|---|---|---|---|---|---|---|---|
| 5.5–6.5 | <0.012 | <0.25 | <0.005 | <0.08 | 3.5–4.5 | <0.15 | <0.04 | Bal. |

After production, the specimens were submitted to two alternative treatments: stress relief heat treatment or HIP treatment. The stress relief heat treatment consisted of slow and controlled heating up to 670 °C, followed by a maintenance period at 670 ± 15 °C for 5 h and, finally, by cooling in air to room temperature. In the HIP treatment the specimens were submitted to a controlled heating up to 920 °C, followed by a maintenance period at 920 ± 15 °C for 2 h in pressured chamber at 100 MPa and cooling in air to room temperature. After the heat treatments, all specimens were submitted to a polishing process in order to reduce the surface roughness ($R_z$ = 2.32 μm) and to have better observation of crack growth in FCG tests.

## 2.2. Metallographic Analysis

Samples were cut from the cross section of the specimens (after post-processing treatments) and prepared following the recommendation of ASTM E3 [21] to study microstructure and hardness. The surfaces were etched with Kroll's reagent (6% $H_2NO_3$, 1% HF, and 93% $H_2O$), as recommended in ASTM E407 [22], and then were observed and photographed in an optical microscope Leica DM 4000 M LED.

Figure 2a shows the microstructure of heat treated (HT) specimens while Figure 2b presents the microstructure obtained for the HIP treated specimens. In both images the darker zones correspond to primary columnar β phase grains while the lighter areas correspond to fine needles of the martensitic phase α. Comparing both figures it is possible to see more dark zones (β phase) and less fine needles of the martensitic phase α (lighter areas), in the case of the HIP treatment (Figure 2b). The higher amount of the β phase in the specimens treated by HIP is due to the higher heating temperature in the HIP treatment (920 °C) that surpasses the transition temperature where the transformation of α phase to β phase occurs (882 °C).

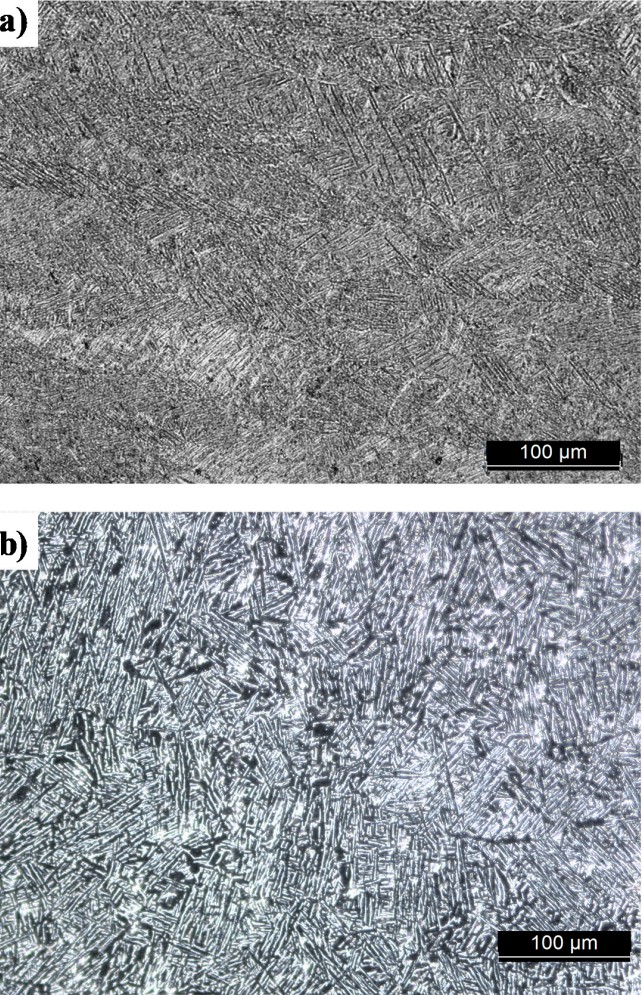

**Figure 2.** Optical metallographics: (**a**) Heat treated specimens and (**b**) HIP treated specimens.

The same samples were used to obtain the Vickers hardness following the ASTM E384-11e1 [23], using a Stuers Duramin 1 hardness tester with a 1 kg of test load and random measurement indentation positions. The HIP treated samples exhibited an average hardness value of 350 HV1 while the HT samples an average value of 405 HV1. The β phase is characterized by a lower hardness than the α

phase. Therefore, the higher transformation of α phase to β phase during the HIP treatment leaded to a loss of hardness of 13.6%.

### 2.3. Tensile Properties

The monotonic tensile tests were performed at room temperature in an Instron mechanical tensile/compression testing machine, model 4206, using a displacement rate of 2 mm/min and the specimen's geometry of Figure 1a. The tensile tests were performed according to the recommendations of ASTM B528-16 [24] standard and using a 12.5 mm gauge length extensometer, Instron 2620-601 (Instron, Norwood, MA, USA), to acquire the elongation.

Table 2 shows the results of the tensile tests for the HIP and HT treated specimens. The loss of hardness described previously also leads to a loss of mechanical resistance in the specimens submitted to HIP treatment and to an increase in ductility of 33.5%. The higher transformation of α phase to β phase that occurs during the HIP treatment, given that the β phase is characterized by having lower hardness and greater ductility than the α phase, leads to the lower resistance and higher ductility obtained in HIP treated specimens.

**Table 2.** Monotonic mechanical properties of the SLM Ti-6Al-4V alloy after post-treatment.

| Sample | $\sigma_{UTS}$ [1] (MPa) | $\sigma_{ys}$ [2] (MPa) | $\varepsilon_f$ [3] (%) | E [4] (GPa) |
|---|---|---|---|---|
| Ti-6Al-4V + HIP | 996 | 951 | 26.4 | 126 |
| Ti-6Al-4V + HT | 1142 | 1106 | 19.6 | 126 |
| % difference (HIP over HT) | −12.8 | −14.0 | +33.5 | 0 |

[1] Ultimate tensile strength; [2] Yield strength; [3] Strain at failure; [4] Young's modulus.

### 2.4. Low Cycle Fatigue Behaviour

Figure 3 shows stress-strain loops obtained with the specimen geometry presented in Figure 1a. The low cycle fatigue tests were carried out in a Dartec servo hydraulic machine (TestResources, Minneapolis, MN, USA), applying a sinusoidal wave under fully-reversed strain-controlled conditions ($R_\varepsilon = -1$) with constant strain rate ($d\varepsilon/dt$) of $8 \times 10^{-3}$ s$^{-1}$, as is recommended in ASTM E606 [1] standard. The load cycles produce material softening, i.e., a progressive reduction of load range. The stress amplitude shows a reduction from the first cycle to the stabilized cycles.

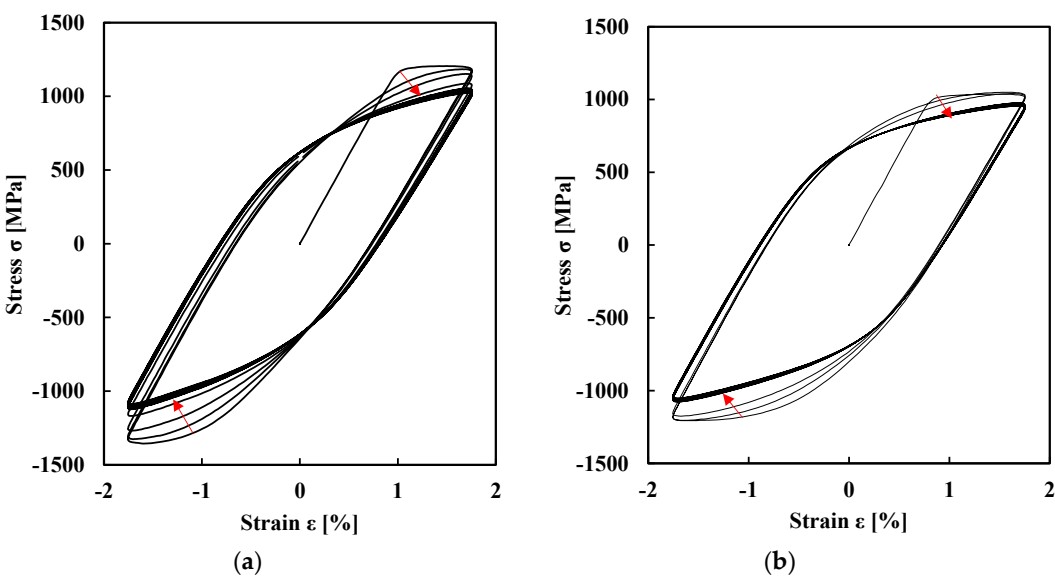

**Figure 3.** Experimental stress-strain loops of SLM Ti-6Al-4V: (**a**) HT treatment; (**b**) HIP treatment. $\varepsilon_a = 1.75\%$.

The Bauschinger effect of both post-treatments can be observed in Figure 3, which is more accentuated in the HIP treated specimens. Additionally, this titanium allow presents some tension-compression asymmetry in the hysteresis loops, where the tensile stress is lower than the compressive stress. This tension-compression asymmetry is higher in the case of the HIP-treated specimens. The analysis of the hysteresis loops also allows to conclude that the HIP-treated specimens presents higher ductility and deformation energy, defined by the largest area within the hysteresis loops. These results are in agreement with the transformation of the microstructure occurring during the HIP treatment and its consequences in the predominance of phase β.

## 2.5. Fatigue Crack Growth Rate

The fatigue crack growth tests were performed according with ASTM E647 standard [20], at room temperature using a 10 kN capacity Instron EletroPuls E10000 machine (Instron, Norwood, MA, USA), under loading control and a stress ratio of $R = 0.05$, at a loading frequency of 10 Hz. The tests were conducted under constant load amplitude, i.e., under increasing $\Delta K$. Table 3 presents the minimum and maximum forces, the initial and final crack lengths and the extreme values of $\Delta K$. The value of $\Delta F = F_{max} - F_{min}$ was increased for the Ti-6Al-4V + HIP to reduce the test duration, increasing the $\Delta K$ at the beginning of the test (Paris regime). Throughout the tests, the crack length was measured using a travelling microscope (45×) with an accuracy of 10 μm.

**Table 3.** Parameters of FCG tests.

| Sample | $F_{min}$ (N) | $F_{max}$ (N) | $a_i$ (mm) | $a_f$ (mm) | $\Delta K_{min}$ (MPa·m$^{0.5}$) | $\Delta K_{max}$ (MPa·m$^{0.5}$) |
|---|---|---|---|---|---|---|
| Ti-6Al-4V + HIP | 132 | 2643 | 6.7 | 25.5 | 9.1 | 49.9 |
| Ti-6Al-4V + HT | 86 | 1711 | 8.8 | 25.3 | 6.9 | 31 |

Figure 4a shows *da/dN*-$\Delta K$ plots in log-log scales for both post-processing treatments. The increase of $\Delta K$ increases the FCG rate significantly, as is well known. The post-processing treatment did not produce a significant influence of FCG rate. There is some difference only for $\Delta K$ higher than 25 MPa·m$^{0.5}$. Note that typically there is a limited effect of microstructure in regime II of crack propagation. The Paris law regime, i.e., a linear variation in log-log scales, is evident at intermediate load ranges. The Paris laws obtained for the HIP treated and HT materials were, respectively:

$$\frac{da}{dN} = 10^{-7.59}(\Delta K)^{2.99} \tag{1}$$

$$\frac{da}{dN} = 10^{-8.11}(\Delta K)^{3.405} \tag{2}$$

where the units of *da/dN* and $\Delta K$ are mm/cycle and MPa·m$^{0.5}$, respectively. The limits of $\Delta K$ considered to define these laws were 11 and 30 MPa·m$^{0.5}$ for the HIP material, and 7 and 30 MPa·m$^{0.5}$ for the heat-treated titanium alloy. A load shedding procedure was followed to obtain fatigue threshold, according the recommendations of ASTM E647 [20]. Threshold values of 4.3 and 3.2 MPa·m$^{0.5}$ were obtained for HIP and heat treatment post-processing, respectively.

Figure 4b compares the FCG rate obtained for the Ti-6Al-4V with those obtained for other metallic alloys [25]. All the results presented were obtained for $R = 0.1$ or $R = 0.05$. The Ti-6Al-4V results are below the curves obtained for the aluminum alloys and above the curves obtained for steels (18Ni300 and 304L stainless steel). The slopes of the curves in the Paris law regime for the different materials are similar.

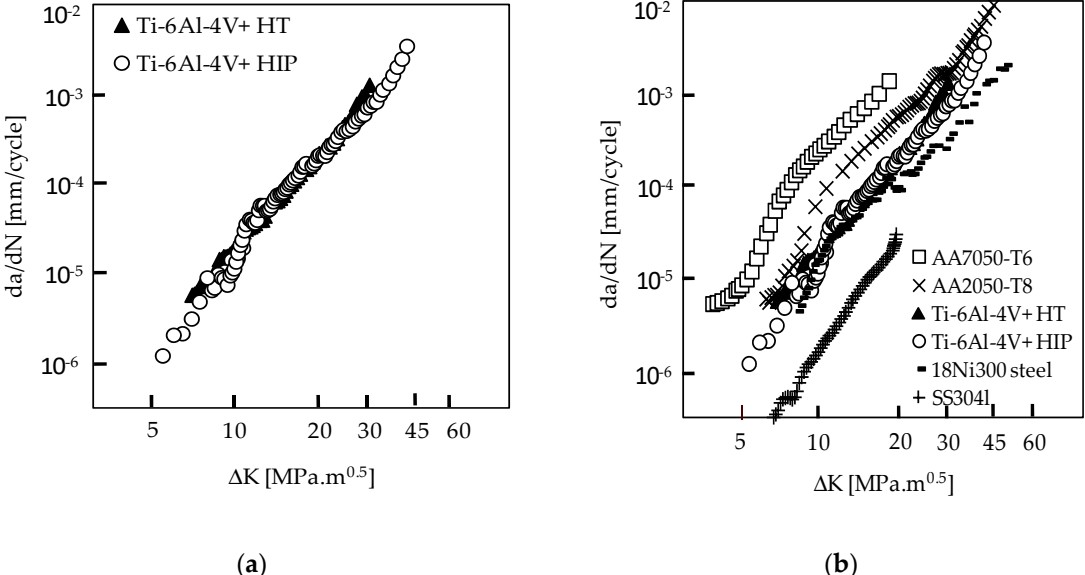

**Figure 4.** Experimental FCG rate: (**a**) Ti-6Al-4V with HIP and HT; (**b**) comparison with other alloys.

## 3. Finite Element Model

The numerical simulations of the fatigue crack growth were performed using the in-house finite element code DD3IMP, which was originally developed to simulate sheet metal forming processes [26]. The adopted numerical model considers the elasto-plastic behavior of the deformable body (CT specimen), assuming the mechanical behavior isotropic. Since the specimen presents geometric, material and loading symmetry relative to the plane of the crack (see Figure 1b), only the upper part of the CT specimen was simulated. The contact of the crack flanks is modeled considering a fixed flat (rigid) surface in the symmetry plane, which prevents the overlapping of crack flanks during unloading. Each finite element mesh of the specimen geometry (upper half part) is composed of approximately 7200 linear hexahedral finite elements, using a selective reduced integration technique [27] to avoid volumetric locking. Only the region near the crack tip is refined to reduce the computational cost, as shown in Figure 5. Although real cracks have a non-zero radius at the tip, the crack tip shape was simplified in the presented finite element model, adopting a sharp crack in all simulations. Thus, the crack is defined by applying adequate boundary conditions to the mesh. The refined element size ($8 \times 8 \; \mu m^2$) in this zone allows the accurate evaluation of the strong stress gradients. This mesh size results from a sensitivity analysis previously performed for a different material [28]. Only a single layer of elements was used in the thickness direction since either plane strain or plane stress conditions are adopted in the numerical model to reduce the computational effort of the numerical solution. Additionally, the specimen thickness was significantly reduced (from 6 mm to 0.1 mm) to obtain plane stress conditions using solid finite elements.

Since the fatigue crack growth rates are usually very low, the numerical simulation of the crack tip advance until fracture requires a very large number of loading cycles, which dictates an enormous computational cost. Therefore, the continuous advance of the crack tip was properly replaced by a set of small crack propagations (<200 μm), uniformly distributed over the crack path. The initial crack sizes adopted in this study were 7 mm, 10 mm, 13 mm, 16 mm, 19 mm, 22 mm, and 24 mm. The finite element mesh is updated accordingly to the initial crack size by shifting the refined region of the mesh, shown in Figure 5.

The cyclic loading is applied in a single point of the specimen hole, as illustrated in Figure 5. Although the geometry of the hole is circular in the experimental specimen (see Figure 1b), a square hole with the same area was created in the numerical model, which simplifies the procedure of mesh generation. Since the thickness of the specimen used in the numerical model is 0.1 mm, which is

different from the one adopted in the experimental work (6 mm), the values of the experimental loads (see Table 3) were divided by 60 to obtain the loads applied in the numerical model. Each load cycle presents a triangular shape with a duration of 2 s (loading + unloading).

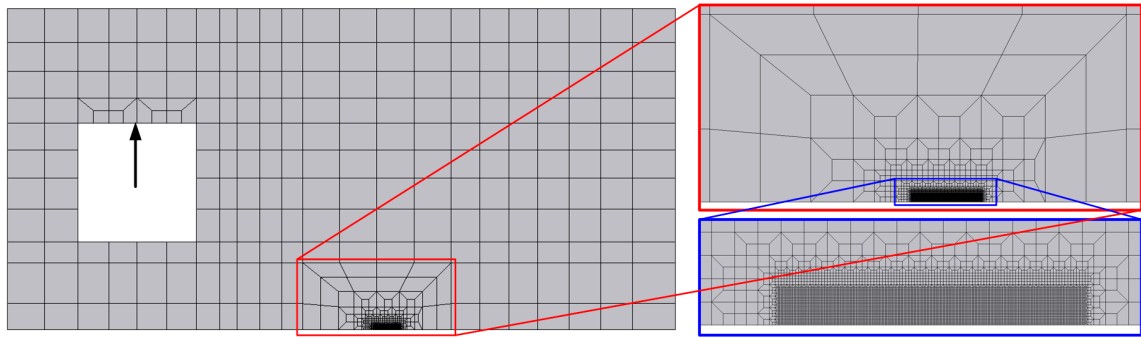

**Figure 5.** Finite element mesh of the CT specimen (upper half part), including the zoom with mesh details to highlight the mesh refinement near the crack tip.

### 3.1. Elasto-Plastic Constitutive Model

In this study, the elasto-plastic behavior of the materials (Ti-6Al-4V with HIP and Ti-6Al-4V with heat treatment) is modeled by phenomenological constitutive models. The elastic behaviour is considered isotropic, which is described by the Hooke's law. The elastic properties are assumed identical for both materials studied. Regarding the plastic response, the isotropy is defined by the von Mises yield criterion while the isotropic hardening behaviour is described by the Swift law, where the flow stress is given by:

$$Y = K(\varepsilon_0 + \bar{\varepsilon}^{\mathrm{P}})^n \quad \text{with} \quad \varepsilon_0 = \left(\frac{Y_0}{K}\right)^{1/n}, \tag{3}$$

where $\bar{\varepsilon}^{\mathrm{P}}$ denotes the equivalent plastic strain, while $K$, $\varepsilon_0$, and $n$ are the isotropic hardening parameters. The kinematic hardening behaviour is described by the Armstrong–Frederick model [29], which can be written as:

$$\dot{\mathbf{X}} = C_X\left(X_{Sat}\frac{\boldsymbol{\sigma}' - \mathbf{X}}{\bar{\sigma}} - \mathbf{X}\right)\dot{\bar{\varepsilon}}^{\mathrm{P}}, \tag{4}$$

where $\boldsymbol{\sigma}'$ is the deviatoric Cauchy stress tensor, $\mathbf{X}$ is the back-stress tensor and $\bar{\sigma}$ is the equivalent stress. $C_X$ and $X_{Sat}$ are the kinematic hardening parameters. In the present study, the material is assumed isotropic and presents tension-compression symmetry. Although the tension-compression asymmetry can be important, the effect is not visible in the experimental stress-strain loops from the low cycle fatigue tests. The crack tip plastic strains are predominantly dominated by tensile stresses. However, the plastic strain at the crack tip is also generated by tensile compression stresses (during the unloading). Therefore, the accurate the accurate modelling of the tension-compression symmetry is important to accurately predict the FCG rate [30].

The isotropic and kinematic hardening parameters were simultaneously calibrated using the stress–strain curves of the experimental low cycle fatigue tests, presented in Figure 3. The parameter's calibration was based on the least-squares minimization of the differences between analytically fitted and experimentally measured values of stress. The obtained hardening parameters are listed in Table 4, for each post-processing treatment analysed.

The comparison between the experimental and numerical stress vs. The accumulated plastic strain curves is presented in Figure 6 for each post-processing treatment studied (Ti-6Al-4V with HIP and Ti-6Al-4V with heat treatment). The adopted constitutive model allows for accurately describing the mechanical behavior of both treatments.

**Table 4.** Material parameters used in Swift isotropic hardening law coupled with the Armstrong–Frederick kinematic hardening law to describe the plastic behaviour of each Ti-6Al-4V.

| Material | $Y_0$ (MPa) | $K$ (MPa) | $n$ | $C_X$ | $X_{Sat}$ (MPa) |
|---|---|---|---|---|---|
| Ti-6Al-4V + HIP | 823.5 | 707.1 | −0.029 | 104.3 | 402.0 |
| Ti-6Al-4V + HT | 700.0 | 738.6 | −0.013 | 88.1 | 585.2 |

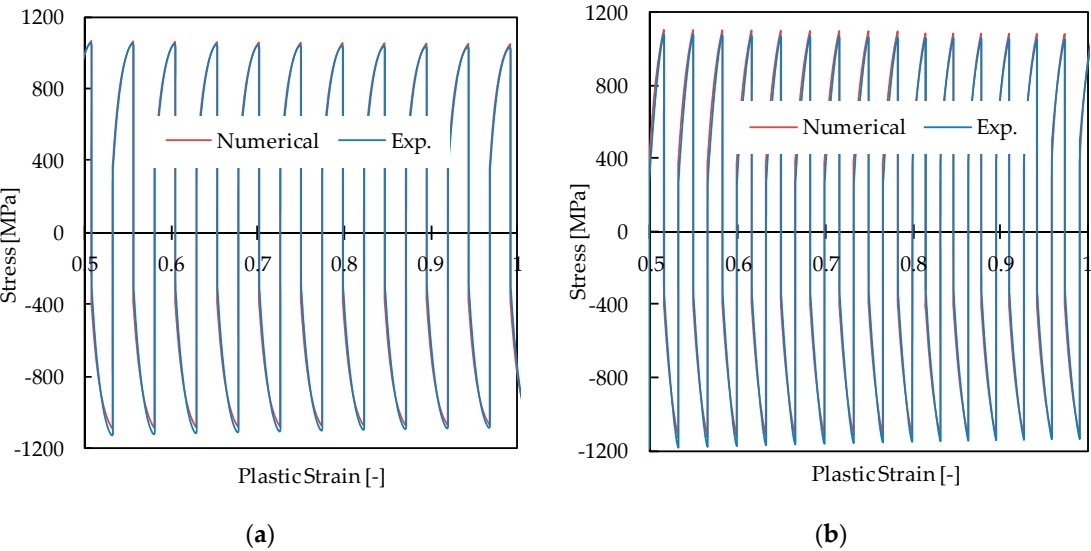

**Figure 6.** Comparison between experimental and predicted stress-strain cycles: (**a**) SLM Ti-6Al-4V with HIP; (**b**) SLM Ti-6Al-4V with heat treatment.

### 3.2. Fatigue Crack Growth Criteria

Considering the specimen geometry and the loading mode, the crack growth direction is known, i.e., the crack path is on the symmetry plane. In the present finite element model, the nodes over the crack path are released according to the proposed fatigue crack growth criterion, allowing for simulating the advance of the crack tip. However, the crack growth is discontinuous due to the discretization, i.e., the crack increment is defined by the finite element size (8 μm) existing in the zone of the crack path.

Two different fatigue crack growth criteria were proposed and compared in this study. Both criteria are based on the plastic strain value at the crack tip, which is assumed to be the main crack driving force. Since the equivalent plastic strain is numerically evaluated only at the Gauss points while the crack propagation is defined by nodal release, the plastic strain at the crack tip (node) is required. Hence, it is obtained by average of the equivalent plastic strain evaluated in the two Gauss points around the crack tip node (immediately behind and ahead). Additionally, the release of the node defining the crack tip occurs always at the instant of minimum load. The numerical value of da/dN is obtained dividing the crack increment of 8 μm (element size) by the number of load cycles required to reach the critical value of plastic strain.

In both presented fatigue crack growth criteria, the crack tip node is released when the plastic strain reaches a critical value. Considering the here called incremental plastic strain (IPS) crack growth criterion, the crack tip node is released when the increment of plastic strain (from the previous release) reaches a critical value [18]. This increment of plastic strain is evaluated from the value existing in this node at the end of the previous release. Thus, the plastic strain accumulated by this node until the previous nodal release is ignored in this fatigue crack growth criterion. On the other hand, the second approach is here called total plastic strain (TPS) crack growth criterion. In this case, the crack tip node is released when the total plastic strain reaches a critical value. Accordingly, this crack growth criterion considers the plastic strain accumulated during the entire cyclic loading. Note that these

two parameters have different concepts behind. The IPS assumes that FCG is due to irreversible deformation acting at the crack tip, while the TPS assumes that the damage accumulation is responsible for FCG. Both proposed criteria require only a single material parameter, which simplifies its usage.

## 4. Numerical Results

### 4.1. Calibration of the Material Parameters

The calibration of the material parameters involved in each proposed crack growth criterion was performed by comparing numerical and experimental FCG rates at a specific value of crack length. In the present study, the critical value of $\Delta\varepsilon^p$ required for the IPS crack growth criterion was evaluated considering two different values of crack length ($a_0 = 7$ mm and $a_0 = 22$ mm). Two extreme crack lengths were considered because it was not clear whether the calibration should be done for a relatively small or large crack length. Therefore, two different values of $\Delta\varepsilon^p$ are obtained, using the experimental FCG rate measured for the crack length $a_0 = 7$ mm and $a_0 = 22$ mm. Additionally, the material parameters calibration is performed independently for plane strain and plane stress conditions since the numerical FCG rates are dependent from the adopted boundary conditions. The thickness of the CT specimens used in the experimental work was 6 mm, which does not guarantee a pure plane stress or plane strain conditions. Therefore, both conditions were tested.

Considering the IPS crack growth criterion, the influence of the critical value of $\Delta\varepsilon^p$ on the predicted FCG rate is presented in Figure 7, using two different values of crack length ($a_0 = 7$ mm and $a_0 = 22$ mm) and comparing plane strain with plane stress conditions. Since the mechanical behavior of the material is influenced by the post treatment, the analysis for Ti-6Al-4V with HIP and Ti-6Al-4V with heat treatment is presented in Figure 7a,b, respectively. Increasing the critical value of $\Delta\varepsilon^p$ leads to a decrease of the predicted FCG rate since the number of loading cycles required to reach the critical value is higher. Hence, the evaluation of the critical value of $\Delta\varepsilon^p$ is obtained when the predicted and experimental FCG rates are coincident. In order to reduce the number of required numerical simulations, either linear interpolation or linear extrapolation has been adopted to find the critical values of $\Delta\varepsilon^p$ for each crack length ($a_0 = 7$ mm and $a_0 = 22$ mm).

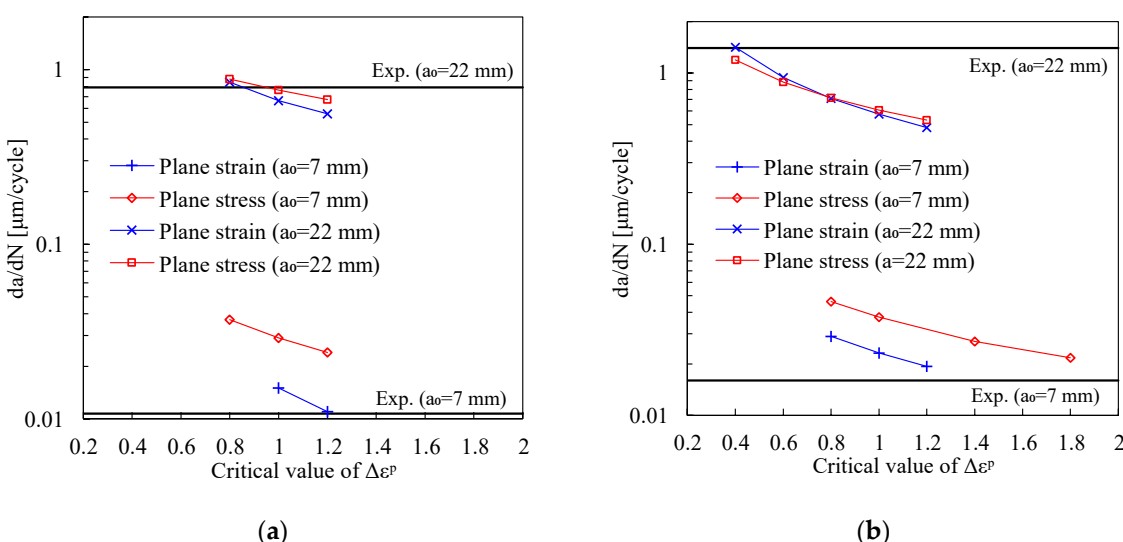

**Figure 7.** Calibration of the critical value of $\Delta\varepsilon^p$ involved in the IPS crack growth criterion, considering two different values of crack length ($a_0 = 7$ mm and $a_0 = 22$ mm) and comparing the plane strain with plane stress conditions: (**a**) SLM Ti-6Al-4V with HIP; (**b**) SLM Ti-6Al-4V with HT.

The critical values of $\Delta\varepsilon^p$ used in the IPS crack growth criterion to describe the mechanical behavior of the Ti-6Al-4V material are listed in Table 5. The calibration was performed for two different values

of crack length ($a_0$ = 7 mm and $a_0$ = 22 mm), considering both plane strain and plane stress conditions in the numerical analysis, as already mentioned. For both post-treatments, the obtained critical value of $\Delta\varepsilon^p$ is lower when the crack length is largest. The discrepancy between plane strain and plane stress conditions is more significant for $a_0$ = 7 mm (see Figure 7), particularly for the Ti-6Al-4V with heat treatment (see Table 5). This material also presents the largest difference of calibrated critical values of $\Delta\varepsilon^p$ using different values of crack length. Finally, the influence of crack length is more important than the influences of post-processing treatment or stress state. Although the experimental *da/dN*-$\Delta K$ curves are similar (regime II) for both post-processing treatments (see Figure 4a), the critical value of $\Delta\varepsilon^p$ is significantly different comparing the post-processing treatments (see Table 5). This is a consequence of the approach used to evaluate the value of $\Delta\varepsilon^p$, matching the numerical and experimental FCG rate in a predefined value of $\Delta K$, which is identical for both post-processing treatments. Considering the largest value of $\Delta K$ used in the calibration procedure ($a_0$ = 22 mm), the corresponding experimental FCG rate is within regime II and III for Ti-6Al-4V with HIP and Ti-6Al-4V with heat treatment, respectively. Critical values of 1.10 and 0.786 were obtained in a previous work [18] for the 2024-T351 aluminum alloy and 18Ni300 steel, respectively, and therefore have the same order of magnitude of the values presented in Table 5.

**Table 5.** Values of $\Delta\varepsilon^p$ for the IPS crack growth criterion. Calibration performed for both post-treatments of the SLM Ti-6Al-4V, using two different values of crack length and comparing plane strain with plane stress conditions.

| Material | Plane Strain | | Plane Stress | |
|---|---|---|---|---|
| | $a_0$ = 7 mm | $a_0$ = 22 mm | $a_0$ = 7 mm | $a_0$ = 22 mm |
| Ti-6Al-4V + HIP | 1.215 | 0.859 | 1.533 | 0.958 |
| Ti-6Al-4V + HT | 1.254 | 0.406 | 2.233 | 0.266 |

Since the size of the plastic wake is very small for the smallest initial crack length ($a_0$ = 7 mm), the difference between the crack growth criteria (IPS and TPS) is negligible when the crack length is relatively small. Thus, the critical values of $\Delta\varepsilon^p$ used in the TPS crack growth criterion are the ones obtained for the IPS crack growth criterion using $a_0$ = 7 mm, which are listed in Table 5 for each post-treatment and both stress state conditions. In fact, using the same set of parameters in both crack growth criteria, the FCG rate predicted by the TPS crack growth criterion is higher than the one obtained with the IPS crack growth criterion. Nevertheless, this difference is negligible for small values of crack length, allowing to use the material parameters previously obtained for the IPS crack growth criterion.

The evolution of the predicted plastic strain at the crack tip is presented in Figure 8, considering plane stress conditions in the numerical analysis of the Ti-6Al-4V with HIP, comparing both crack growth criteria (IPS and TPS). Although only two nodal propagations are presented, different values of initial crack length are evaluated, ranging from $a_0$ = 10 mm up to $a_0$ = 22 mm. Considering the smallest initial crack length ($a_0$ = 10 mm), the predictions obtained from different crack growth criteria are identical (compare Figure 8a,b). On the other hand, the impact of the selected crack growth criterion on the predicted plastic stain evolution is substantial for the largest initial crack length ($a_0$ = 22 mm). Considering the IPS crack growth criterion, the crack tip node is released when the increment of plastic strain (from the previous release) reaches the critical value ($\Delta\varepsilon^p$ = 1.533), as shown in Figure 8a. On the other hand, using the TPS crack growth criterion, the crack tip node is released when the total plastic strain reaches the critical value ($\Delta\varepsilon^p$ = 1.533), as shown in Figure 8b.

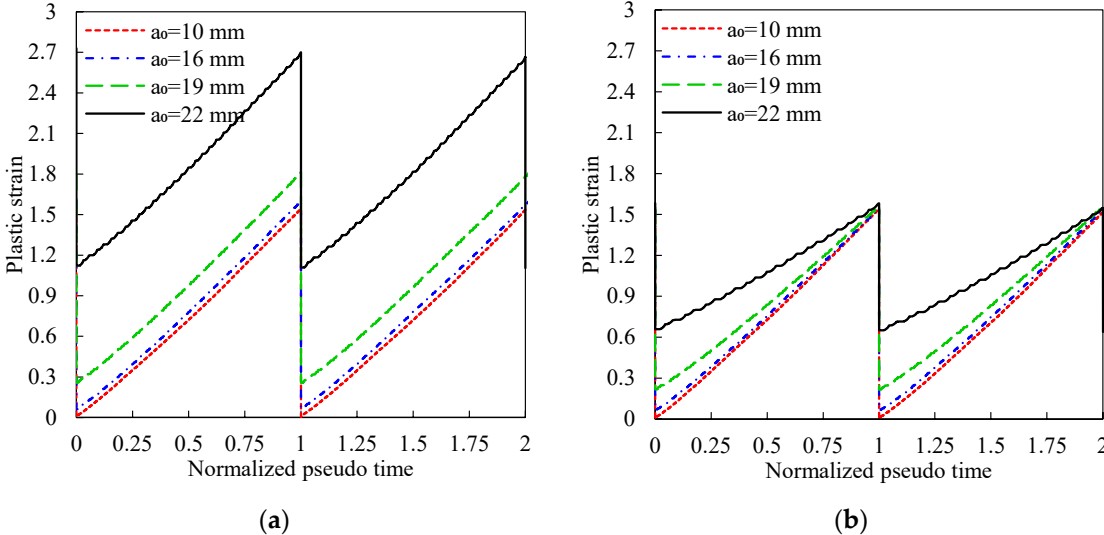

**Figure 8.** Evolution (two nodal propagations) of plastic strain in the crack tip, predicted under plane stress conditions for different values of initial crack length in the SLM Ti-6Al-4V with HIP: (**a**) IPS crack growth criterion; (**b**) TPS crack growth criterion.

### 4.2. Stabilization of the Plastic Wake Zone

The predicted FCG rate for each initial crack length can present a transient effect at the beginning of the crack propagation. This is related with the stabilization of cyclic plastic deformation and the formation of residual plastic wake. Considering the TPS crack growth criterion, the predicted FCG rate as a function of the applied loading cycles is presented in Figure 9 for different values of the initial crack length, comparing the post-treatments (Ti-6Al-4V with HIP and Ti-6Al-4V with heat treatment). Despite the differences in the mechanical behavior of both materials (see Table 2), the predicted behavior in terms of FCG rate evolution is similar for both materials. Since the FCG rate is considerably higher for larger values of the initial crack length, the number of loading cycles applied between node releases is lower. Therefore, the total number of loading cycles applied in the numerical simulation is significantly lower for largest values of crack length, as highlighted in Figure 9. Despite this, the predicted length of propagation is large due to the higher FCG rate. Therefore, the simulated crack propagation is larger than 100 μm for $a_0$ = 24 mm and lower than 35 μm for $a_0$ = 7 mm. The stabilization of the predicted FCG rate is slower when plane stress conditions are assumed. Indeed, considering plane strain conditions in the numerical simulations, the stabilization of the FCG rate is very fast, as shown in Figure 9a for Ti-6Al-4V with HIP and Figure 9c for Ti-6Al-4V with heat treatment. The propagation required to stabilize *da/dN* is linked with crack closure phenomenon, which affects the effective load range. This phenomenon is more relevant for plane stress state, which explains the longer stabilization distances. Since the size of the plastic zone is higher for the largest values of the crack length, the stabilization of the FCG rate requires more crack propagations for $a_0$ = 24 mm. The number of markers in Figure 9 defines the amount of crack propagations simulated in each case). Since the FCG rate small is lower for small values of the crack length, more loading cycles are required for the same crack length propagation.

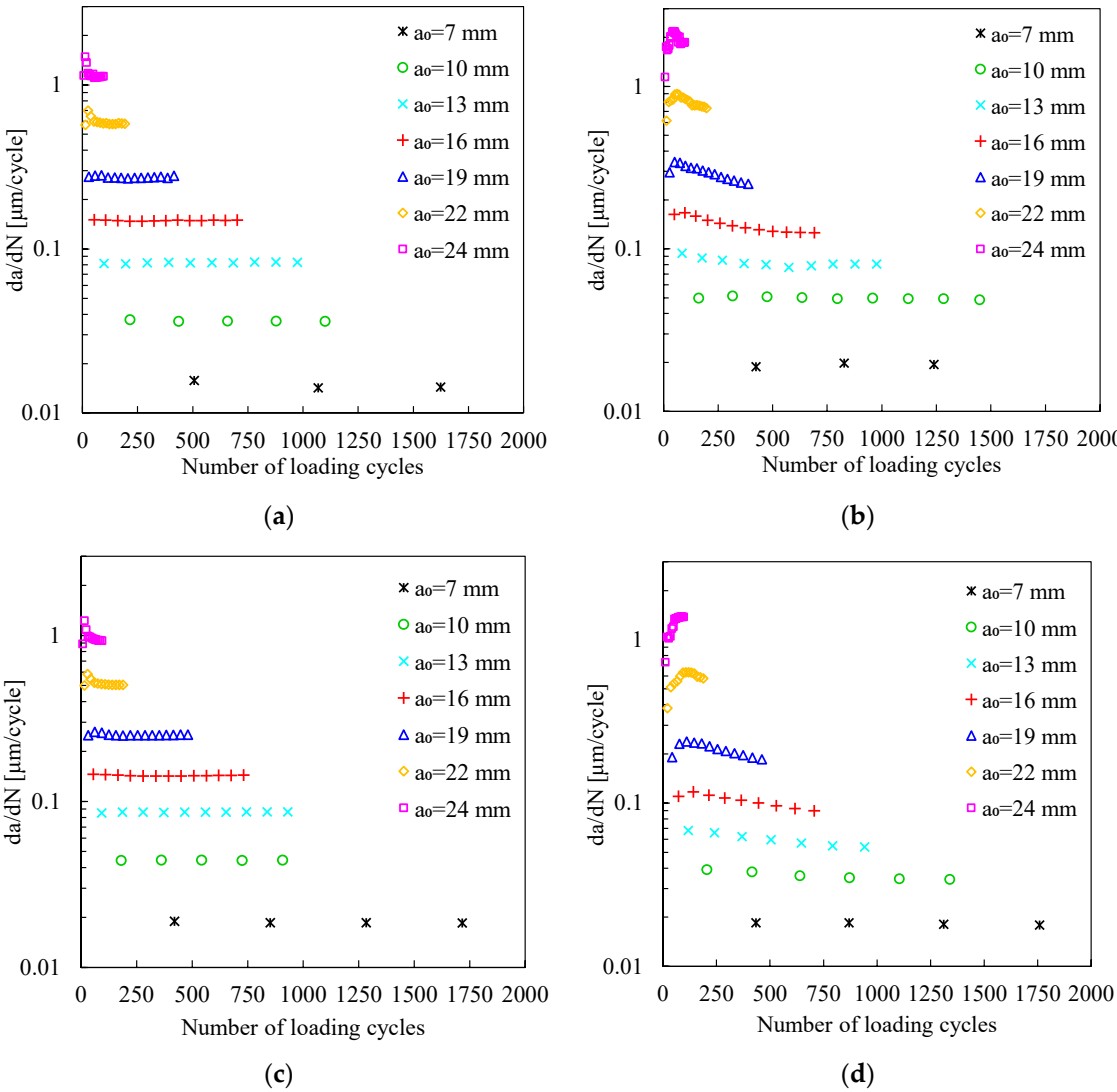

**Figure 9.** Predicted fatigue crack growth rate as a function of the applied loading cycles for different values of initial crack length: (**a**) SLM Ti-6Al-4V with HIP under plane strain conditions using the TPS criterion with the critical value $\Delta\varepsilon^p = 1.215$; (**b**) SLM Ti-6Al-4V with HIP under plane conditions using the TPS criterion with the critical value $\Delta\varepsilon^p = 1.533$; (**c**) SLM Ti-6Al-4V with HT under plane strain conditions using the TPS criterion with the critical value $\Delta\varepsilon^p = 1.254$; (**d**) SLM Ti-6Al-4V with HT under plane stress conditions using the TPS criterion with the critical value $\Delta\varepsilon^p = 2.233$.

*4.3. Fatigue Crack Growth Rate*

The comparison between experimental and predicted *da/dN-ΔK* curves for Ti-6Al-4V with HIP is presented in Figure 10, separating the numerical results obtained under plane strain conditions from those obtained under plane stress conditions. The numerical values of FCG rate presented in this section were evaluated in stable zone of the FCG rate, i.e., after the initial transient regime. Globally, the numerical results are in good agreement with the experimental ones. Considering the IPS crack growth criterion, the increase of the critical values of $\Delta\varepsilon^p$ adopted in the numerical simulations leads to a shift down of the *da/dN-ΔK* curve, as highlighted in Figure 10. In other words, the change of the critical value of cumulative plastic strain produces a vertical translation of *da/dN-ΔK* curve. Regarding the TPS crack growth criterion, considering small values of stress intensity factor range ($\Delta K < 12$ MPa·m$^{0.5}$), the numerical predictions of FCG rate are identical to those obtained with the IPS crack growth criterion. On the other hand, for large values of stress intensity factor range, the FCG

rate predicted by the TPS crack growth criterion is higher than that predicted by the IPS crack growth criterion. This behavior is in agreement with the predicted plastic strain evolution at the crack tip, i.e., the difference between the two crack growth criteria is visible only for large crack lengths (see Figure 8). Therefore, the accuracy of the predicted FCG rate is globally improved by using the TPS crack growth criterion. Adopting the TPS crack growth criterion, for small values of stress intensity factor range ($\Delta K < 12$ MPa·m$^{0.5}$) the difference between experimental and predicted $da/dN$-$\Delta K$ curves is lower considering plane strain conditions (see Figure 10a), while for large values of stress intensity factor range ($\Delta K > 24$ MPa·m$^{0.5}$) the difference is lower considering the plane stress conditions (see Figure 10b).

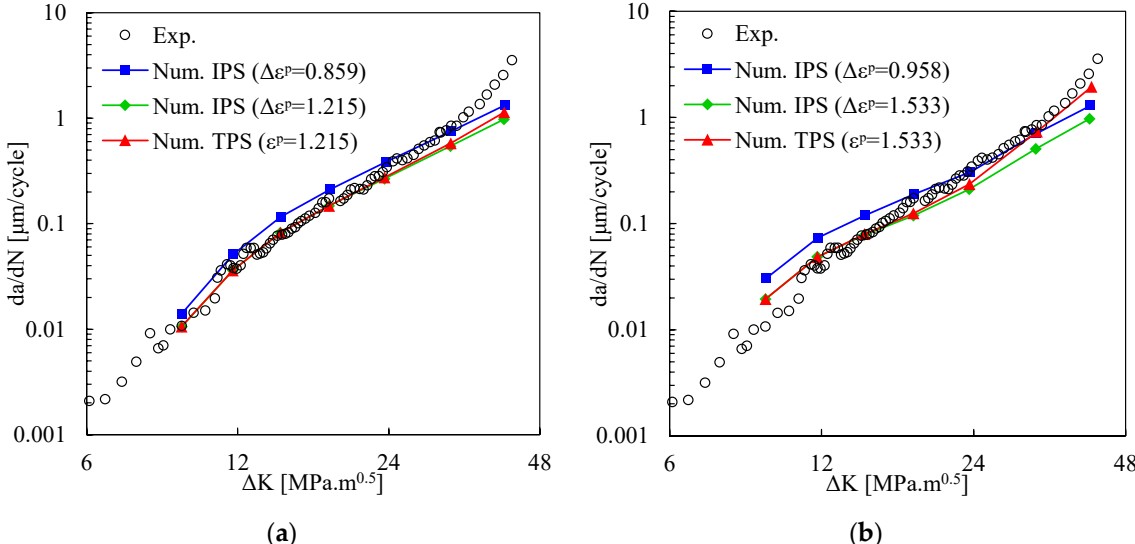

**Figure 10.** Comparison between experimental and predicted *da/dN*-Δ*K* curves for SLM Ti-6Al-4V with HIP: (**a**) FCG rates predicted under plane strain conditions using both the IPS and the TPS criteria; (**b**) FCG rates predicted under plane stress conditions using both the IPS and the TPS criteria.

Figure 11 presents the comparison between experimental and predicted *da/dN*-Δ*K* curves for Ti-6Al-4V with heat treatment, using both plane strain and plane stress conditions in the numerical simulations. Once again, the numerical results are globally in agreement with the experimental ones, particularly for low values of Δ*K*. Adopting the IPS crack growth criterion, the predicted *da/dN*-Δ*K* curve shifts down when the critical value of $\Delta\varepsilon^p$ increases, as highlighted in Figure 11 for both plane strain and plane stress conditions. This variation is large for Ti-6Al-4V with heat treatment in comparison with the Ti-6Al-4V with HIP (Figure 10) since the range used for the critical value of $\Delta\varepsilon^p$ is higher in this post-treatment (see Table 5). Since the critical values of $\Delta\varepsilon^p$ were calibrated independently for $a_0 = 7$ mm and $a_0 = 22$ mm (see Table 5), the numerical and experimental *da/dN*-Δ*K* curves are coincident in the point used to define the critical value of $\Delta\varepsilon^p$.

Considering small values of stress intensity factor range ($\Delta K < 12$ MPa·m$^{0.5}$), the difference between IPS and TPS crack growth criteria is negligible, as highlighted in Figure 11. Nevertheless, for large values of stress intensity factor range ($\Delta K > 24$ MPa·m$^{0.5}$), the slope of the predicted *da/dN*-Δ*K* curve increases when the TPS crack growth criterion is adopted, particularly under plane stress conditions (see Figure 11b). Indeed, for this range of crack lengths, only the TPS crack growth criterion under plane stress conditions provides an accurate prediction of the experimental *da/dN*-Δ*K* slope. On the other hand, for small crack lengths, the numerical simulation under plane strain conditions allows to obtain an accurate prediction for the slope of the *da/dN*-Δ*K* curve.

The distribution of the plastic strain near the crack tip is presented in Figure 12 for different values of crack length, adopting the TPS crack growth criterion applied to the Ti-6Al-4V with heat treatment. The size of the plastic zone increases with the increase of the crack length, as could be

expected. Additionally, the size of the plastic zone predicted under plane stress conditions is larger than assuming plane strain conditions in the numerical simulations. This can be related with the calibration of the critical value of $\Delta\varepsilon^p$ used in the TPS crack growth criterion, i.e., the critical value of $\Delta\varepsilon^p$ used in the simulations under plane stress conditions is significantly higher than the one adopted in the plane strain conditions (see Table 5). The stabilization of the predicted plastic wake zone is quick for small crack lengths (see Figure 12), which is agreement with the evolution of the FCG rate presented in Figure 9. Considering the largest initial crack length ($a_0$ = 22 mm), the size of the plastic zone ahead of the crack tip is significantly higher using plane stress conditions than adopting plane strain conditions in the simulation, as highlighted in Figure 12. This yields an increase of the FCG rate predicted by the TPS crack growth criterion in comparison with the IPS criterion (see Figure 11b).

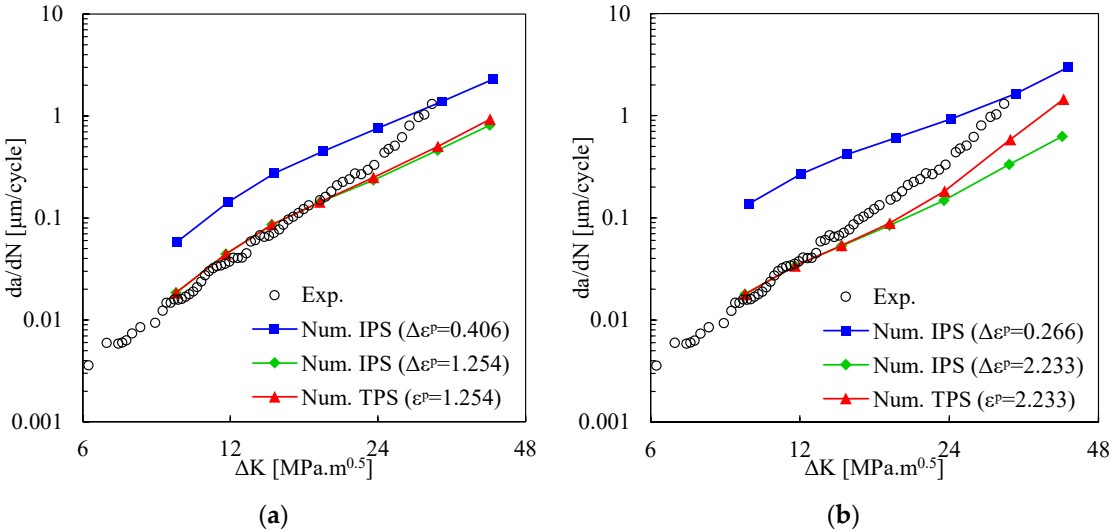

**Figure 11.** Comparison between experimental and predicted *da/dN-ΔK* curves for SLM Ti-6Al-4V with HT: (**a**) Fatigue crack growth rates predicted under plane strain conditions using both the IPS and the TPS criteria; (**b**) fatigue crack growth rates predicted under plane stress conditions using both the IPS and the TPS criteria.

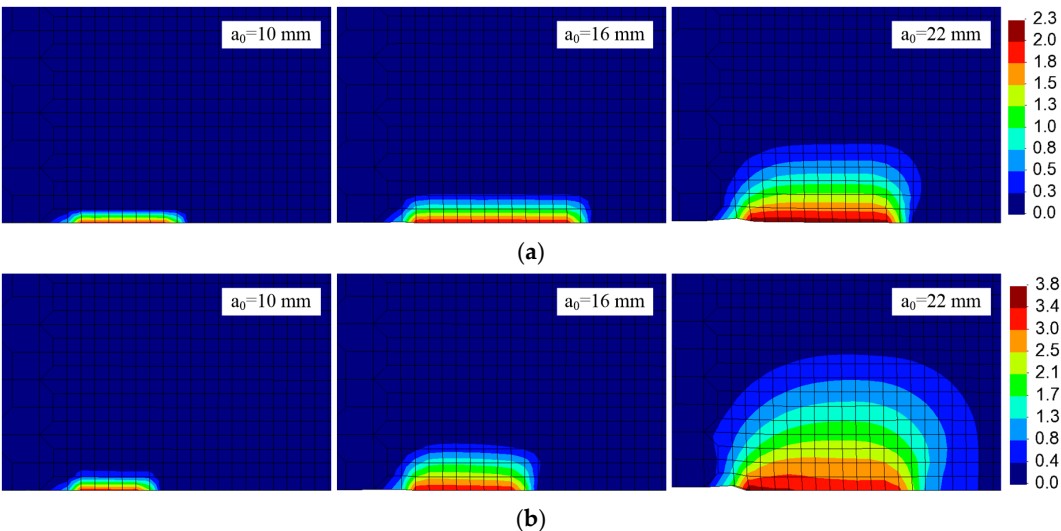

**Figure 12.** Plastic strain distribution near the crack tip for different values of initial crack length in the SLM Ti-6Al-4V with heat treatment: (**a**) Plane strain conditions using the TPS criterion with the critical value $\Delta\varepsilon^p$ = 1.254; (**b**) plane stress conditions using the TPS criterion with the critical value $\Delta\varepsilon^p$ = 2.233.

## 4.4. Effect of Sstress Ratio

The effect of the stress ratio on the predicted FCG rate is evaluated for the material Ti-6Al-4V with HIP, considering the TPS crack growth criterion (with the critical value $\Delta\varepsilon^p = 1.533$) under plane stress conditions. Figure 13a presents the predicted $da/dN$-$\Delta K$ curve for $R = 0.05$ and $R = 0.5$. The increase of the stress ratio was carried by increasing both the minimum and the maximum load values, maintaining constant the loading amplitude. Considering the contact of crack flanks, the increase of $R$ produces an increase of $da/dN$, which is a typical result. However, without contact of crack flanks there is almost no influence of stress ratio. This indicates that cyclic plastic deformation is not responsible for the effect of stress ratio. Therefore, crack closure is the main responsible for the effect of stress ratio, assuming that cyclic plastic deformation is the crack driving force for FCG. A similar trend was obtained for the 2024-T351 aluminum alloy [18]. Anyway, there is a small increase for the highest values of $\Delta K$ (see Figure 13a).

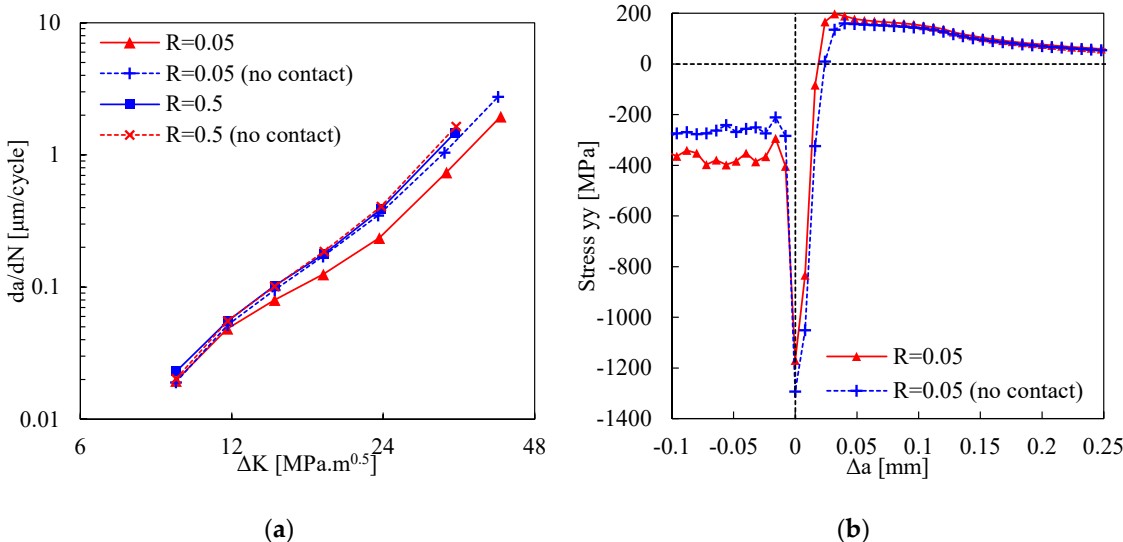

**Figure 13.** Effect of stress ratio on $da/dN$-$\Delta K$ curve for SLM Ti-6Al-4V with HIP using the TPS criterion under plane stress conditions: (**a**) Influence of the contact between crack flanks for two different values stress ratio; (**b**) influence of the contact between crack flanks on the predicted stress ahead of crack tip evaluated at minimum load for $R = 0.05$.

Figure 13b presents the stress ahead of crack tip predicted at the instant of minimum load for $a_0 = 16$ mm of crack length and $R = 0.05$, where $\Delta a = 0$ defines the location of the crack tip. The stress is compressive immediately ahead of crack tip, presenting the highest magnitude at the crack tip. The compressive stresses are in a region with dimension lower than 0.03 mm. For locations away from the crack tip the predicted stress is tensile, as shown in Figure 13b. The influence of the contact between crack flanks seems insignificant. However, the magnitude of the compressive stress immediately ahead of the crack tip is higher when the contact between crack flanks is ignored. Additionally, the dimension of the zone under compressive stress is also higher for this condition (see Figure 13b). Thus, the geometrical constraints imposed by the contact between crack flanks leads to relief the compressive stresses at the instant of the minimum load.

## 4.5. Fatigue Threshold

The numerical model was also used to predict the value of fatigue threshold. It is assumed that fatigue crack propagation only occurs if the plastic deformation at the crack tip is increasing. Thus, there is a maximum value of $\Delta K$ below of which the plastic deformation at the crack tip is not increasing (crack propagation does not exist). Since the loading occurs at constant amplitude

(minimum and maximum force identical in all cycles), a single loading cycle without an increase of plastic deformation is enough to define the fatigue threshold. In order to reduce $\Delta K$, the minimum load was gradually increased, while the maximum load was kept constant. Figure 14a plots the variation of the FCG rate with $\Delta K$. These values were obtained without contact of crack flanks, which is very interesting since crack closure phenomenon has a major effect on the values of FCG rate. Load shedding strategies, used in the experimental work to reduce crack closure effects, can therefore be avoided here. The decrease of $\Delta K$ reduces *da/dN*, as expected, and for $\Delta K$ lower than 7.6 MPa·m$^{0.5}$ there is no crack growth at all. The extrapolation to da/dN = 0 gives a predicted fatigue threshold $\Delta K_{th}$ = 7.3 MPa·m$^{0.5}$, which is expected to be valid in vacuum. The experimental value obtained here for the Ti-4Al-6V was 3.1 MPa·m$^{0.5}$. Similar values were found in literature for the same material. Moshier et al. [31] obtained values of $\Delta K_{th}$ of 4.6 and 2.9 MPa·m$^{0.5}$, for $R$ = 0.1 and $R$ = 0.5, respectively. Oguma and Nakamura [11] obtained a value of 4 MPa·m$^{0.5}$ in air for $R$ = 0.1. Therefore, there is a significant difference between the numerical prediction and the experimental value. In fact, a factor of two was proposed to exist between the value in vacuum and the effective fatigue threshold in air [32]. This difference was attributed to a different damage mechanism of FCG at relatively low load ranges, which is associated with environmental effect at the crack tip [33].

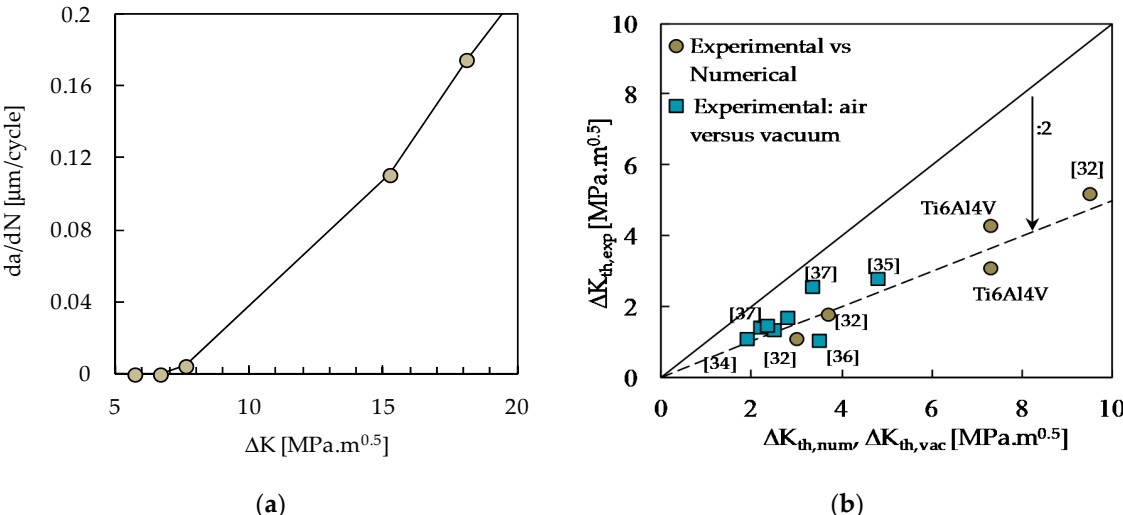

**Figure 14.** (**a**) *da/dN* versus $\Delta K$ near threshold; (**b**) experimental and numerical results of fatigue threshold [32,34–37].

Figure 14b presents experimental values of fatigue threshold versus numerical predictions [32,34–37]. The squares indicate experimental results from literature, obtained in air and vacuum. There is a reasonable agreement with the dashed line, which represents a factor of about two between numerical and experimental results of the effective fatigue threshold. The points corresponding to Ti-6Al-4V are also close to the dashed line. Note that the prediction of fatigue threshold using a continuum mechanics approach is not consensual. Different authors state that the onset of FCG must be studied using models based on the analysis of dislocation movement at the crack tip, and that it is not possible to study fatigue threshold with continuum mechanics models [38]. They assumed that the crack progresses by shear mechanisms along the critical slip system with maximum resolved shear stress [39]. The advancing crack may be arrested at a grain level obstacle such as a grain or a twin boundary. The application of a continuum mechanics model is expected to be more problematic for coarse-grained materials.

## 5. Discussion

An analysis was developed to check if the small-scale yielding (SSY) assumption is fulfilled or not. This analysis is based on CTOD measured at the first node behind crack tip, i.e., at a distance of

8 µm. In a previous work it was found that SYY assumption is valid if $\delta_e/\delta_t > 75\%$, i.e., if the elastic CTOD, $\delta_e$, corresponds to at least 75% of the total CTOD, $\delta_t$ [40]. On the hand, large-scale yielding (LSY) is supposed to occur when $\delta_e/\delta_t < 60\%$, i.e., if the elastic CTOD is less than 60% of the total CTOD. Figure 15 presents $\delta_e/\delta_t$ parameter versus $\Delta K$. As can be seen, there is a linear decrease of elastic deformation with the increase of $\Delta K$ and, therefore, with the increase of crack length. For the largest crack lengths, the SSY assumption is no longer valid, although the Ti-6Al-4V is a material with high yield stress. This reinforces the need of the use on non-linear parameters, instead of $\Delta K$, in the study of FCG, even for this high strength material.

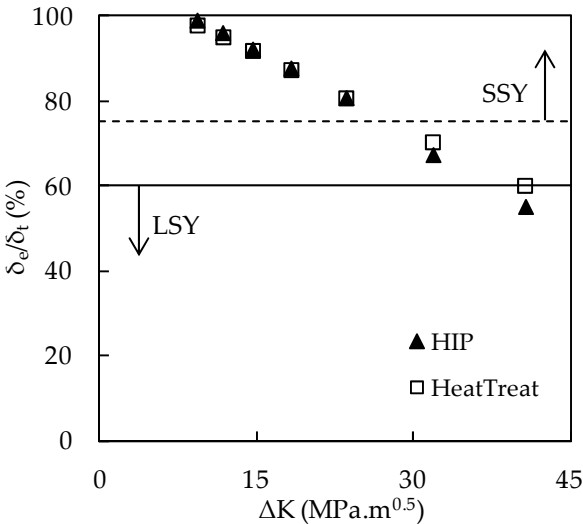

**Figure 15.** Variation of $\delta_e/\delta_t$ with $\Delta K$ ($F_{min} = 2.2$ N; $F_{max} = 44.05$ N; plane strain).

Another main issue is the identification of the mechanisms responsible for FCG. The reasonable agreement between the numerical predictions and experimental results (particularly for the TPS criterion) indicates that cyclic plastic deformation is probably the main mechanism. Figure 16 presents experimental results and different possible positions for the numerical curve, whose position depends on the point used for calibration of critical cumulative plastic strain. At position A, the numerical predictions are above the experimental results, and the difference increases with the reduction of $\Delta K$. Oguma and Nakamura [11] and Yoshinaka et al. [12] showed than the effect of air increases $da/dN$, particularly at low values of $\Delta K$. Therefore, the numerical curve, which is obtained assuming that cyclic plastic deformation is the only driving force, is on the wrong side of the experimental results. Alternatively, this curve may be placed at position B, i.e., below the experimental results. However, in this case there is a coincidence of results at the lowest value of $\Delta K$, indicating that there is no effect of the environment. Therefore, the numerical curve must probably be placed at position C. Now, the difference observed at low values of $\Delta K$ may be explained by the effect of the environment. At relatively high values of $\Delta K$, there is a significant difference, which must be explained by mechanisms activated by the maximum load of the cycle. Kujawski [41] proposed a fatigue crack driving force parameter, $(K_{max})^\alpha (\Delta K^+)^{1-\alpha}$. A $\alpha$ value of 1 was proposed for granite, while a value of 0.5 was proposed for 2024-T351 and 7075-T6 aluminium alloys, indicating that there is a significant influence of $K_{max}$ even for ductile materials.

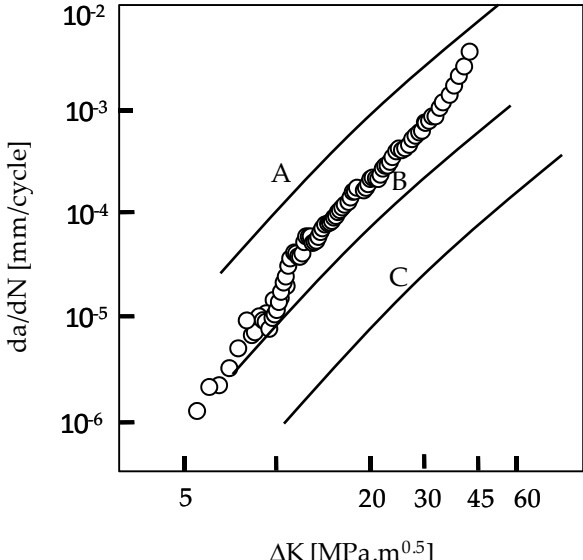

**Figure 16.** Possible positions of the numerical *da/dN-ΔK* curve (lines A, B, and C) relative to the experimental results.

## 6. Conclusions

Fatigue crack growth in Ti-6Al-4V alloy was studied using the classical experimental approach based on ΔK and a more evolved approach based on crack tip plastic strain. Cylindrical specimens and CT specimens were produced by SLM and submitted to post-processing treatments of HIP or heat treatment. The cylindrical specimens were submitted to low cycle fatigue tests under fully-reversed strain-controlled conditions, in order to obtain stress-strain loops which were subsequently used to fit an elastic-plastic constitutive material model. The CT specimens were used to obtain classical *da/dN-ΔK* curves. The main conclusions of the experimental work are:

- The post-processing treatments produced changes in microstructure, hardness and tensile properties. The HIP treatment produced more transformation of martensitic α-phase to β-phase. Since the β-phase is softer than the α-phase, the hardness of HIP treated material is lower than the heat treated material, while the ductility is higher.
- On the other hand, the post-processing treatment produced a limited effect on *da/dN-ΔK* curves.

A numerical model was developed to predict FCG, assuming that crack tip plastic deformation is the crack driving force. Two different FCG criteria are proposed: the incremental plastic strain (IPS) criterion and the total plastic strain (TPS) criterion. IPS assumes that FCG is due to irreversible deformation acting at the crack tip, while TPS assumes that the damage accumulation is responsible for FCG. The critical value of plastic strain required to release nodes, i.e., to propagate the crack, was adjusted using experimental values of da/dN. Different values of crack length were considered, namely the extreme values of crack length (7 and 22 mm), for plane stress and plane strain states. The numerical predictions of *da/dN* were obtained dividing the crack increment of 8 μm (element size) by the number of load cycles required to reach the critical value of plastic strain. This critical value was calibrated using experimental values of *da/dN*. Different values were proposed considering extreme values of crack length (7 and 22 mm), for plane stress and plane strain states. The main conclusions of the numerical work are:

- A transient behavior was found for *da/dN* at the beginning of the numerical crack propagation, which is linked with the stabilization of cyclic plastic deformation and particularly with the formation of residual plastic wake responsible for crack closure phenomenon. The extension of this transient increases with the initial crack length and is more relevant for the plane stress state.

- A good agreement was found between stabilized numerical predictions and experimental results, indicating that cyclic plastic deformation is the main mechanism responsible for crack propagation. The accuracy of the predicted FCG rate is globally improved by using the TPS crack growth criterion, instead of the IPS criterion. Note that the reasoning behind these two concepts is substantially different, deserving a particular attention in future studies.
- Neglecting the contact of crack flanks there is almost no influence of stress ratio on the *da/dN-ΔK* curve, which indicates that cyclic plastic deformation is not responsible for R effects. An extrinsic phenomenon, i.e., the contact of crack flanks, is needed to explain R effects.
- The SSY assumption was found to be invalid for the largest crack lengths studied, reinforcing the need of using non-linear parameters to study FCG. Additionally, the numerical approach based on a non-linear parameter provides a better understanding and discussion of crack tip phenomena, even in the high strength material studied here.

**Author Contributions:** Conceptualization: D.M.N.; methodology: F.V.A.; software: D.M.N.; validation: J.S.J.; formal analysis: F.F.F. and J.S.J.; writing—original draft preparation: F.V.A., J.S.J., and D.M.N.; writing—review and editing: P.A.P. All authors have read and agreed to the published version of the manuscript.

**Funding:** This research was funded by projects with reference PTDC/EME-EME/28789/2017 and PTDC/EMEEME/ 31657/2017, financed by the European Regional Development Fund (FEDER), through the Portugal-2020 program (PT2020), under the Regional Operational Program of the Center (CENTRO-01-0145-FEDER-028789 and CENTRO-01-0145-FEDER-031657) and the Foundation for Science and Technology IP/MCTES through national funds (PIDDAC). This research is also sponsored by FEDER funds through the program COMPETE—Programa Operacional Factores de Competitividade- and by national funds through FCT—Fundação para a Ciência e a Tecnologia-, under the project UIDB/00285/2020.

**Conflicts of Interest:** The authors declare no conflict of interest. The funders had no role in the design of the study; in the collection, analyses, or interpretation of data; in the writing of the manuscript; or in the decision to publish the results.

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
