# Peer review of "Numerical Prediction of the Fatigue Crack Growth Rate in SLM Ti-6Al-4V Based on Crack Tip Plastic Strain"

_metals, doi:10.3390/met10091133_

Round 1

Reviewer 1 Report

Great work! See attached document for minor comments/recommendations.

Reviewer 2 Report

-

Reviewer 3 Report

Excellent manuscript that is both very interesting and well prepared. Despite this there remains a few minor improvements that can be made to make the manuscript more comprehensive in nature. 

Line 111: What was the surface roughness after the polishing process (although the reader can assume that both HIP and HT samples are largely similar, the crack initiation phenomenon is strongly governed by this factor). 

Table 2: It seems as though the HIP treatment provides a large significant increase in ductility at the sacrifice of strength, as the proportion of alpha and beta phases change. Is there no grain refining mechanism imposed on the microstructure due to the large pressure induced during the HIP process? Is there any residual stress and/or twins which remain after the specimens cool following HIP?

Table 3: The delta k threshold and Max for the HIP specimens are significantly higher than the HT specimens (presumably from the higher ductility). If the HIP specimens have a lower strength, and both conditions exhibit cyclic material softening... then why were the load amplitudes selected for HIP higher than those for HT? In Table 2 HT was presented first, and HIP second, this can be confusing to a careless reader who doesn't read the sample ID properly and assumes it is the same as above. 

Figure 5: At the threshold of mesh refinement (when you go from quad brick elements to more Trapezoidal shaped elements, is the jacobian and element quality of certain corner elements still sufficient for crack tip FEM?

Figure 7: What were the specifics of the CT specimen geometry that were used for plane stress and plane strain?

Figure 7: It seems as though the HT specimens now have a higher critical value for delta epsilon plastic, despite having a lower threshold and maximum delta k? Can the authors comment on how both of these quantities can be related to the increase in ductility observed in the HIP treated specimen (compared with the HT specimen)? 

Figure 12: How was the effects of anisotropy and asymmetry that are common in wrought forms of Titanium handled using the swift hardening criterion that was adopted? Is this not a concern with SLM Titanium components as they are more isotropic in nature and have a lower propensity to twin in compression? Or is it not vitally important for modelling the crack tip plastic strains as they are predominantly dominated by tensile damage and plastic strain (as opposed to the bulk material)?

Round 2

Reviewer 2 Report

-
